# New mechanism to identify cost savings in English NHS prescribing: minimising 'price per unit', a cross-sectional study

Richard Croker, Alex J Walker, Seb Bacon, Helen J Curtis, Lisa French, Ben Goldacre

## ABSTRACT

**Background** Minimising prescription costs while maintaining quality is a core element of delivering high-value healthcare. There are various strategies to achieve savings, but almost no research to date on determining the most effective approach. We describe a new method of identifying potential savings due to large national variations in drug cost, including variation in generic drug cost, and compare these with potential savings from an established method (generic prescribing).

**Methods** We used English National Health Service (NHS) Digital prescribing data, from October 2015 to September 2016. Potential cost savings were calculated by determining the price per unit (eg, pill, millilitre) for each drug and dose within each general practice. This was compared against the same cost for the practice at the lowest cost decile to determine achievable savings. We compared these price-per-unit savings to the savings possible from generic switching and determined the chemicals with the highest savings nationally. A senior pharmacist manually assessed whether a random sample of savings were practically achievable.

**Results** We identified a theoretical maximum of £410 million of savings over 12 months. £273 million of these savings were for individual prescribing changes worth over £50 per practice per month (mean annual saving £33 433 per practice); this compares favourably with generic switching, where only £35 million of achievable savings were identified. The biggest savings nationally were on glucose blood testing reagents (£12 million), fluticasone propionate (£9 million) and venlafaxine (£8 million). Approximately half of all savings were deemed practically achievable.

**Discussion** We have developed a new method to identify and enable large potential cost savings within NHS community prescribing. Given the current pressures on the NHS, it is vital that these potential savings are realised. Our tool enabling doctors to achieve these savings is now launched in pilot form at OpenPrescribing.net. However, savings could potentially be achieved more simply through national policy change.

Centre for Evidence Based Medicine, Nuffield Department of Primary Care Health Sciences, University of Oxford, Oxford, UK

**Correspondence to**
Dr Ben Goldacre;
ben.goldacre@phc.ox.ac.uk

## Strengths and limitations of this study

- ► The novel method of making prescribing savings described here can be directly implemented with the help of our associated tool, which is updated monthly.
- ► We were able to measure the potential savings across all prescribing in England, eliminating bias. We also removed seasonal variation by aggregating savings over 12 months.
- ► The method described uses an automated method to identify potential savings, meaning that clinical judgement must be used to determine where switching between presentations is appropriate.
- ► Some of the identified savings may not be achievable due to complexities in prescribing practices and unmeasurable factors such as rebates.

National Health Service (NHS) in England, it is therefore increasingly important that savings are found where possible. However, there is relatively little in the academic literature comparing methods for optimising costs in prescribing. Therapeutic switching is one conventional approach to achieve savings, where patients are switched to cheaper treatments from the same class. It is somewhat complex to implement, as it requires clinical expertise and knowledge of comparative effectiveness; and the change may not be suitable for all patients.[2] Generic switching is a common and more straightforward approach to saving resources; patients are switched from branded drugs to cheaper generic alternatives that are chemically identical.[3 4]

There has been an overall increase in generic prescribing over the last decade, with 84.1% of prescriptions in England prescribed generically in 2015, compared with 80.1% in 2005.[1] Moreover, between 1972 and 2013 the proportion changed from 20% to 84%, saving an estimated £7.1 billion for the NHS, meaning that costs have not increased in proportion with the increase in prescription numbers.[5] This existing high level of generic

## INTRODUCTION

The spend on prescribing in England in primary care was £9.3 billion in 2015 and has been broadly increasing in recent years.[1] Given the increasing cost pressures on the

### Box 1 The Drug Tariff and potential for cost savings

In the National Health Service in England, general practitioners prescribe on an FP10 form, which patients then have dispensed at a community pharmacy (or in some rural areas in the surgery itself). The Drug Tariff[6] outlines the reimbursement cost for the majority of medicines prescribed in primary care. Savings by minimising price per unit are possible through various routes:

#### Variation at point of prescribing:
► Switching from brand to generic: there are still a number of prescriptions for conventionally branded preparations where a cheaper generic is available.
► Switching to cheapest 'branded generic': some generic prescriptions have their reimbursement price based on the originator brand, despite there now being specific brands of generic (branded generics) available at lower cost.
► Different formulations: there can be multiple formulations of the same chemical entity on the market (such as capsules and tablets) at different prices, while some have different clinical benefits, others are effectively interchangeable.

#### Variation at point of dispensing:
► Drugs not listed in the Drug Tariff (NP8): some drugs are not listed in Part VIIIA of the Drug Tariff, and therefore pharmacies will be reimbursed at the invoiced cost. This has led to some instances of very large variation in costs depending on which pharmacy dispensed the prescription. Individually formulated and imported medicines: some drugs or formulations are not available as licensed products in England. In these cases, an individually formulated (specials) product may be obtained from a specialist manufacturer. Part VIIIB of the Drug Tariff lists the reimbursement price for many of these which may vary significantly between similar formulations (eg, suspension and solution). Where there is no price listed, the pharmacies will be reimbursed at the invoiced cost. The same applies for imported medicines. This can lead to large variation in costs as per NP8 drugs.
► Different pack sizes: for some medicines such as emollients there may be multiple pack sizes available. In these cases, the reimbursement is dependent on what pack size the pharmacy has endorsed, regardless of what the prescription states, for example, 5×100 g may cost more than 1×500 g for a prescription for 500 g.

### Box 2 Definitions for elements in UK National Health Service (NHS) prescribing data

The following terms are used for UK NHS prescribing data in general, and in this paper:
► An 'item' is a prescription issued by a doctor or other prescriber.
► A 'chemical' is the active ingredient; for example, 'tramadol hydrochloride'.
► The 'formulation' is the form in which the chemical is given; for example, 'tablet', 'capsule', 'liquid' or 'cream'.
► A 'presentation' is all of: the chemical, the strength, the formulation and then the generic name (if only a generic has been prescribed), or the brand name (if a specific brand has been explicitly prescribed). For example, 'tramadol hydrochloride sustained release (SR) 100 mg capsules' or 'Zamadol SR 100 mg capsules'.
► A 'generic-equivalent presentation' is one step higher in the hierarchy than 'presentation': it is the chemical, the dose and the formulation, but not the specific brand. For example, the 'generic-equivalent presentation' of 'tramadol hydrochloride SR 100 mg capsules' would include everything prescribed as the generic 'tramadol hydrochloride SR 100 mg capsules', but also everything prescribed as the brand 'Zamadol SR 100 mg capsules' which is a branded form of 'tramadol hydrochloride SR 100 mg capsules'.
► Every prescription is for a 'quantity' of the 'units' of the treatment; for example, this can be the number of tablets or capsules, the number of injections or inhalers, millilitres of a liquid, or grams of a cream.
► The 'price per unit' is the cost paid by the NHS for each 'unit'.

prescribing means a reduction in the remaining opportunities available for generic switching. However, there is still wide variation in the unit cost of a number of medicines prescribed across England, due to the way the reimbursement system is structured. As a consequence of this variation, the cost to the NHS of a prescription for the same treatment at the same dose can vary widely between practices, depending on the specific presentation that is dispensed; for example, a branded or generic version of the same treatment may have different prices; but different specific 'brands' of 'branded generic' may also have different prices. More details are given in box 1 for the policy and administrative background to these potential savings; and precise definitions of terminology are given in box 2.

As part of the OpenPrescribing.net project, we run an openly accessible service to identify cost-saving opportunities in NHS primary care prescribing data. We set out to develop a method to automatically identify cost-saving opportunities from variation in the price per unit (PPU) of a given treatment, by identifying the PPU in each practice for each dose of each treatment; comparing this against the PPU in the best 10% of most efficient prescribers and using the volume of each treatment prescribed in each practice to rank and prioritise savings opportunities. This is then used to generate a tool which advises practices and Clinical Commissioning Groups (CCGs) on their biggest potential cost savings from switching prescriptions to a different brand or formulation. We then set out to determine the overall cost savings available to the NHS in England through this method, and compare it to cost savings from the current comparable approach of simple generic switching. Importantly, neither method involves switching between different drugs, making both more readily achievable. We also discuss the practical implications of this proposed new approach to cost savings.

## METHODS
### Data
We used data from our OpenPrescribing.net project, which imports prescribing data from the monthly prescribing data files published by NHS Digital.[7] These contain data on cost and volume prescribed for each drug, dose and preparation, for each English general practice. Each row of data within the dataset describes prescribing of a presentation for one practice for that month giving total cost, total number of items (prescriptions) and

total quantity prescribed (see box 2 for terminology). For example, a given practice might have a number of rows of data for tramadol hydrochloride 100 mg modified release (MR) preparations: one for tramadol hydrochloride 100 mg MR tablets, prescribed generically; one for tramadol hydrochloride 100 mg MR capsules, prescribed generically; but also separate rows for, for example, Tramulief SR 100 mg tablets and Zamadol SR 100 mg capsules where these were specified by the prescriber as branded generic presentations. We used 12 months of data, from October 2015 to September 2016.

### General principles

We intend the savings illustrated here to be realistically achievable by a well-implemented medicines optimisation programme, and assume that perfect prescribing is not always possible. We have therefore not used perfect prescribing as a reference to determine potential savings, but instead compared each practice against the performance of the practice at the 10th percentile of best performance for each cost-saving opportunity. We have also assumed that prescribing behaviour changes yielding only very small savings are not necessarily cost-effective. Savings under £1 per practice per month were therefore excluded from all analyses; and for some of the analyses we have applied an additional floor, for example, requiring that each action will save at least £50 per practice per month. For efficient practices already performing at better than the 10th percentile on a given measure (where worsening performance to match the 10th percentile would have resulted in increased costs) possible savings were assumed to be £0. Savings were calculated separately for each month, then aggregated over the year, as individual prices and prescribing may change on a monthly basis. We have also assessed achievability of all savings opportunities by manually reviewing a representative random sample with a senior medicines optimisation pharmacist as described below.

### Calculating savings from 'generic switching'

We calculated savings available due to generic switching by matching each branded drug to the equivalent generic presentation, where available. Following the NHS Digital prescribing data definitions, branded drugs were identified as those with anything other than 'AA' in characters 10 and 11 of the British National Formulary (BNF) code.[8] Their generic equivalents were matched by identifying presentations with the same chemical code (first 9 characters) and format code (last 4 characters), but with 'AA' in characters 10 and 11. Maximum theoretical savings were calculated at the practice level by determining what the cost of prescribing for this treatment and dose would have been if all branded medications had been prescribed at the average cost for the generic equivalent. We then report the level of saving that would have been achieved if each practice prescribed the same proportion of branded drug as the practice at the best performing 10th percentile for this proportion. For the main analysis, we only included positive savings; we also describe how the savings would be affected if situations where generic switching results in increased costs are included.

### Calculating savings from 'PPU' switching

For every individual month, and for every practice, we calculated the mean PPU for every generic-equivalent presentation. For example, this would be the mean PPU of all 'tramadol hydrochloride 100 mg MR capsules' prescribed, regardless of whether this was prescribed as 'tramadol hydrochloride 100 mg MR capsules', or 'Zamadol SR 100 mg capsules' or 'Tramquel SR 100 mg capsules' (each a branded presentation of tramadol hydrochloride 100 mg MR capsules). Generic-equivalent presentations were matched to the code of the generic presentation by collapsing all presentations with the same chemical code (first 9 characters), where characters 14 and 15 match those of characters 12 and 13 of the generic presentation's code, onto the generic presentation's code, to make a generic-equivalent presentation.

Having ascertained the mean PPU in each individual practice for each generic-equivalent presentation, we then identified the practice at the 10th percentile for PPU for each generic-equivalent presentation. We used this price, and the quantity prescribed at each practice to calculate how much each practice could have saved if it had prescribed that generic-equivalent presentation as cost efficiently as the practice at the 10th percentile. Additionally, we have combined all formulations (eg, tablets and capsules) at 'generic-equivalent presentation' level where we are aware that these are clinically interchangeable, as per the table in (online supplementary appendix A). We have also excluded some potential substitutions where it was determined that switches were not comparable, as per (online supplementary appendix B).

While this data processing is complex to describe in full reproducible detail, for end-users (specifically, general practitioners (GPs)) the message is simple: 'tramadol hydrochloride 100 mg MR is available in many forms; they are all interchangeable; here are the cost-saving opportunities from switching.'

### Describing variation

The savings for generic switching and minimising PPU were calculated for each practice, for every generic-equivalent presentation, for each month. We have presented the total savings available nationally from each method; and the number of distinct actions required to obtain those savings, where an action is a practice changing their choice of prescribed presentation for one generic-equivalent presentation (eg, following our notification of potential cost savings, the practice may decide to: 'always prescribe Zamadol 200 mg MR capsules when you want tramadol 300 mg MR, as these are the cheapest').

We provide summary statistics on the size of the cost-saving opportunities. As well as total possible savings, we present savings available if only actions over a certain amount per practice per month were included: these are

presented for hard thresholds (over £50, £100, £500 and £1000) and also represented graphically using continuous thresholds.

We have also calculated the total aggregated savings and actions at each practice by each method, presented summary statistics to describe these overall savings per practice. Lastly, we present aggregated national savings at chemical level, to determine which chemicals offer the greatest level of potential savings, and estimate the savings opportunities from only targeting a specific smaller number of chemicals.

### Achievability

Not all potential savings can be realised. There may sometimes be clinical justification for a specific patient to be prescribed a branded treatment in place of the generic equivalent. Changing between brands (or from brand to generic) may cause patient concern and may possibly either alter adherence or be switched back to the original prescription. Similarly, not all variation in PPU can be addressed by individual clinicians: there may be problems with availability of a specific cheaper branded generic; some variation may be due to pack size or 'specials'. Lastly, some of the money lost in the price paid for a dispensed presentation may be made up through a complex system of 'rebates' paid by specific pharmaceutical companies to specific CCGs on specific products. These arrangements are not routinely disclosed: they, therefore, undermine transparency around price paid by the NHS for medical treatments, and render assessments of inefficiency complex; they also have complex long-term consequences, as they may result in patients being initiated on expensive products long term with an initial discount that is then taken away over time.

To assess the impact of these issues on the savings identified, a senior pharmacist (RC) running a medicines optimisation team at a large CCG manually reviewed the top 10 cost-savings opportunities identified from PPU in 10 randomly selected practices, and categorised them according to their achievability. We also categorised

savings according to whether they arose as a result of 'specials', variation in broken pack size or different areas of the Drug Tariff.

## RESULTS

### Savings from generic and PPU switching

Summary data on national savings are presented in table 1. If every practice substituted equivalent generics at the level of the best performing decile for each presentation, the theoretical maximum saving is £56.3 million, from 1.83 million distinct cost-saving actions. Restricting the analysis to only actions that can save a practice more than £50 per month yields a total of £34.8 million in savings from 298 000 actions, with a median saving of £82 per action. If every practice minimised PPU to the same degree as the best decile of practices for each presentation, then the theoretical annual maximum saving is £410 million, from over 14 million actions. Restricting the analysis to only actions that can save a practice more than £50 per month yields a total of £273 million in savings from optimising PPU, spread across 2.04 million actions, a median cost saving for each practice of £92 per action. The savings from optimising PPU are therefore an order of magnitude greater than those from the conventional approach of generic switching. This is due to a larger number of cost-saving actions available from optimising PPU.

The level of total possible savings varies according to the minimum cost-saving threshold imposed on the data. Savings from limiting actions to only those over higher value thresholds are presented in table 1 for discrete categories, and as continuous data in figure 1, to help guide choices on the trade-off between the savings that can be yielded and the effort required to achieve them.

### Cost savings per practice, generic switching

The mean cost saving possible per practice over the year from generic switching was £6880, across the 8180 practices included; this fell to £4251 when only counting

**Table 1** Potential savings for the two cost-saving methods

| | Generic switching | | | Price per unit | | |
|---|---|---|---|---|---|---|
| | Total annual savings (millions) | No of actions | Median monthly cost saving per action | Total annual savings (millions) | No of actions | Median monthly cost saving per action |
| Theoretical maximum savings | £56.3 | 1 828 802 | £13 | £410.4 | 14 274 013 | £8 |
| Savings over £50/month | £34.8 | 298 094 | £82 | £273.5 | 2 035 124 | £92 |
| Savings over £100/month | £21.9 | 112 701 | £150 | £193.7 | 905 352 | £159 |
| Savings over £500/month | £2.5 | 3167 | £636 | £35.0 | 41 362 | £655 |
| Savings over £1000/month | £0.7 | 476 | £1311 | £12.7 | 7283 | £1342 |

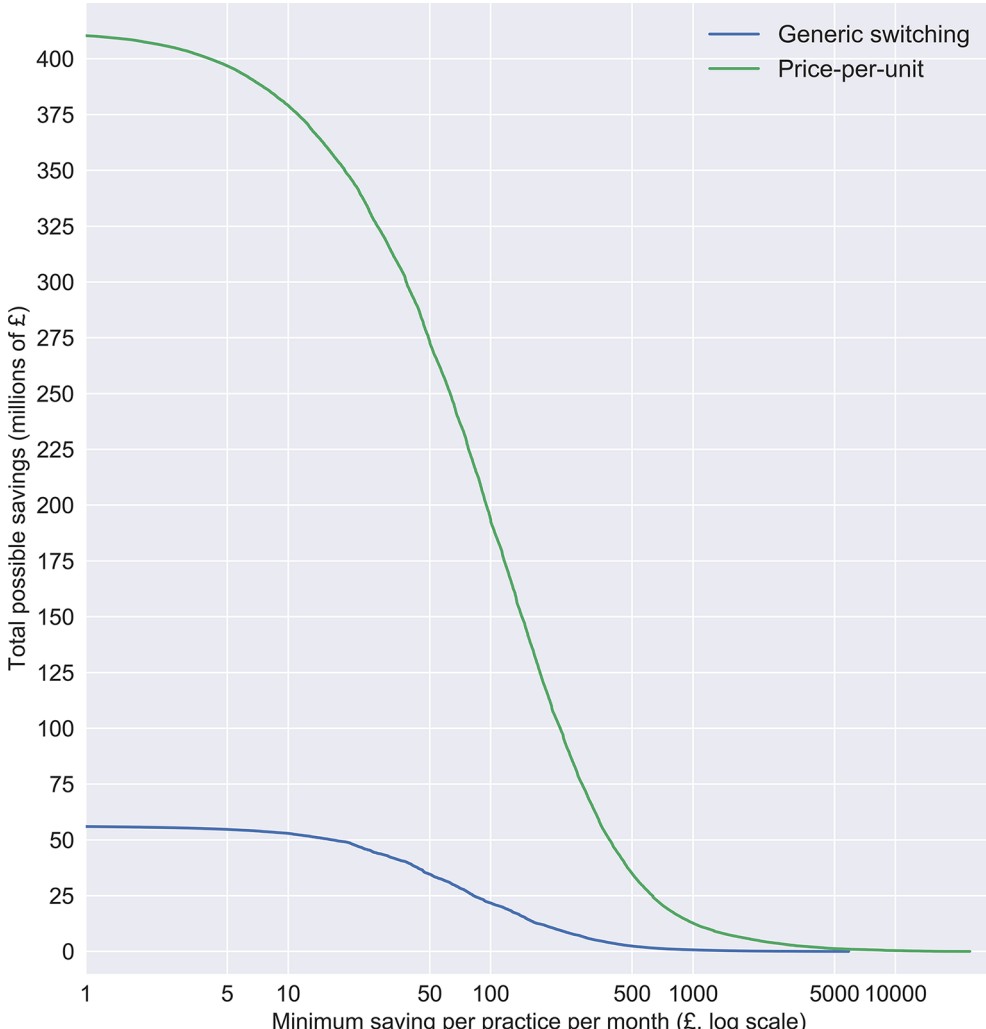

**Figure 1** Distribution of total possible savings (y-axis), showing how much total saving there would be if only those over £x per practice per month (x-axis) were implemented, for both generic switching and minimising price-per-unit.

actions that yield over £50 per month. Practices had a mean of 36 cost-saving actions over £50 per month over the course of the year (median 26, 5th percentile 0, 95th percentile 109). Eight hundred and ten practices (9.9%) had no cost-saving actions over £50 per month.

If only the top 25% of practices made all possible savings over £50 per action, then there would still be £22.5 million of savings from generic switching, and £155.9 million of savings from addressing savings from improvements on PPU. Lastly, we calculate that implementing just the top three savings per practice for each month would yield savings of £23.9 million.

### Cost savings per practice, PPU switching

At the practice level, there was a mean total of £50 166 of savings per practice per year, across the 8180 practices (£33 433 for savings over £50). Practices had a mean of 248 cost-saving actions over £50 per month, over the course of the year (median 209, 5th percentile 0, 95th percentile 620). Four hundred and forty-one practices (5.4%) had no cost-saving actions over £50. If only the least efficient 25% of practices made all possible savings over £50 per

action, then this would still yield £155.9 million of savings. Implementing just the top three savings per practice for each month would yield savings of £86.7 million.

### Cost savings per chemical, generic switching

The cost savings available from generic switching were divided between 578 BNF presentations and 317 different BNF chemicals, with the maximum potential saving at chemical level being for levetiracetam (£10.2 million). The top 10 savings at BNF chemical level are shown in table 2. National savings from addressing generic switching on these chemicals alone would total £30.0 million.

### Cost savings per chemical, PPU switching

The savings from optimising PPU were spread across 3275 BNF presentations and 912 different BNF chemicals, with the maximum potential saving at chemical level being for glucose blood testing reagents (£12.0 million). The top 10 savings at BNF chemical level are shown in table 3. Savings from optimising PPU on these 10 chemicals alone would total £66.5 million. As a proportion of total national efficiency opportunities, cost-saving actions

**Table 2** Top 10 savings for generic switching by British National Formulary (BNF) chemical name

| BNF chemical code | Chemical name | Potential saving per year (£) |
|---|---|---|
| 0408010A0 | Levetiracetam | 10 214 151 |
| 0703021Q0 | Desogestrel | 4 967 839 |
| 0212000B0 | Atorvastatin | 2 546 116 |
| 0301020S0 | Glycopyrronium bromide | 2 480 091 |
| 0106040M0 | Macrogol 3350 | 2 030 447 |
| 0103050E0 | Esomeprazole | 1 925 282 |
| 1106000L0 | Latanoprost | 1 565 643 |
| 0802010M0 | Mycophenolate mofetil | 1 515 336 |
| 0407041T0 | Sumatriptan succinate | 1 403 414 |
| 040801050 | Topiramate | 1 348 505 |

from PPU are dispersed over a wider number of chemicals and a larger number of high-value actions. Yielding these savings therefore requires, to a greater extent, that each practice has access to actionable data on their own specific cost-saving opportunities, rather than a list of most common chemicals to examine for possible savings.

### Achievability of savings

A senior pharmacist running a medicines optimisation team at a large CCG (RC) manually reviewed the top 10 cost-savings opportunities identified from PPU switching in 10 randomly selected practices, and categorised them according to whether they could be achieved by GPs in the NHS. Twelve per cent of savings by cash value were

**Table 3** Top 10 savings for price-per-unit switching by British National Formulary (BNF) chemical name

| BNF chemical code | Chemical name | Potential saving per year (£) |
|---|---|---|
| 0601060D0 | Glucose blood testing reagents | 11 969 900 |
| 0302000N0 | Fluticasone propionate (Inh) | 9 279 710 |
| 0403040W0 | Venlafaxine | 7 699 071 |
| 0302000K0 | Budesonide | 6 631 387 |
| 0408010A0 | Levetiracetam | 5 624 539 |
| 0407010F0 | Co-codamol (codeine phosphate/paracetamol) | 5 203 179 |
| 0105010B0 | Mesalazine (systemic) | 5 190 272 |
| 0704020N0 | Tolterodine | 5 013 993 |
| 0408010H0 | Lamotrigine | 4 996 309 |
| 0302000K0 | Budesonide | 4 926 375 |

regarded as 'very hard to achieve', mostly due to price variation arising from variation in pack size. Certain items, such as creams and emollients, have lower costs per unit for larger pack sizes; for example, if a GP prescribes 2500 g of Aveeno cream, the pharmacist can dispense and endorse 25×100 g packs or 5×500 g packs, with the latter incurring higher costs; this variation is hard for prescribers to control, as it can occur even where the GP specifies 5×500 g. Thirty-eight per cent of savings by cash value were regarded as 'unclear': these were principally treatments where the pharmacist again has extensive discretion, specifically bespoke 'specials' where price can be arbitrarily higher or lower; and drugs which are not covered by the standard NHS Drug Tariff (listed as 'NP8' in the Drug Tariff documentation). Eleven per cent were regarded as 'achievable with additional intervention': this included, for example, savings from different brands of glucose test strips where the switch would also require procurement of a new metre to match the new brand.

There is also an issue of 'primary care rebate schemes'. These are payments made by pharmaceutical companies to CCGs to reimburse them for a proportion of the list price of the medicine, and to possibly incentivise use of specific medicines. While they may help to reduce short-term treatment costs, they may also result in patients being maintained medium term or long term on more expensive interventions and may normalise the use of higher cost medicines across the health service. Rebates are not well known in the medical community, and full details are not routinely disclosed, as the schemes are exempt from Freedom of Information requests due to confidentiality clauses in the contracts between the parties. From reviewing cases where rebates from the pharmaceutical industry have been disclosed by CCGs in response to Freedom of Information Act requests and linking these to prescribing data, we estimate that approximately 7% of all potential PPU savings may be affected, although the medium-term impact on NHS expenditure is inherently hard to model.[9]

It is not possible to use data to automatically identify all PPU savings opportunities that harder to achieve; and achievability for the same savings opportunity will vary regionally depending on how local services are organised. We therefore caution that the savings figures given in this paper should be regarded as estimates, and suggest that an appropriate discount is applied for achievability using the estimates given above, perhaps using an estimated discount of 50%.

### DISCUSSION
### Summary
Our PPU method found a theoretical maximum annual saving of £410 million, compared with £56 million for generic switching. Restricting the analysis to only include those prescribing changes which save a practice more than £50 per month reduced the savings to £274 million from PPU and £35 million from generic switching.

Applying a further discount of 50% for achievability to the PPU savings leaves estimated savings of £137 million. The practically achievable savings from improving PPU efficiency therefore represent 1.5% of the overall NHS spend on primary care prescribing (£9.3 billion in 2015).[1] We also found that in the current pricing market, blindly prescribing generically can sometimes result in increased cost for some drugs, meaning the total potential saving from this older conventional method is further reduced.

## Advantages/disadvantages

We were able to measure all prescribing across the whole of England, meaning that there was no possibility of obtaining a biased sample. Aggregating savings over 12 months removed all seasonal variation. We did not attempt here to compare the savings identified with those for therapeutic switching, another established method of making savings; this was determined to be impractical as therapeutic switching involves switching between similar but distinct drugs, requiring specific clinical judgement in each case, including potentially adjusting dose or other medication; this would additionally require a manually curated list of equivalent treatments covering all treatments prescribed, which is impractical. We are not aware of any detailed analysis of current cost savings from therapeutic switching; however, we note that the savings estimates from older crude estimates[3 4] in commentary papers would no longer hold, as they estimate savings from therapeutic switching in drug classes such as statins where nearly all drug patents expired some time ago.

It is possible that there are additional challenges to achievability of some switches, beyond those described above. For example, rarely there may be licensing differences, where two drugs which are bioequivalent are not both licensed for all possible uses and therefore cannot be used interchangeably by a clinician who is concerned by this discrepancy; rarely there is non-bioequivalence, where some drugs within a generic class cannot always be considered clinically equivalent, and individually manufactured and imported drugs can vary wildly in costs due to different import routes, which are outside the control of the prescriber.

Our calculations use the NHS net ingredient cost (NIC), which is the list price as defined by the Drug Tariff or where agreed with manufacturers, as opposed to the actual cost, which is calculated using the following formula:

Actual cost=(NIC less discount[a])
+payment for consumables[b]
+payment for containers[c]
+out-of-pocket expenses[d]

where a denotes the discount which pharmacies are assumed to have been given by their suppliers (around 7% during the study); b denotes for example, 5 mL spoon (paid at 1.24 p for all prescriptions, not just those requiring a consumable); c denotes the original pack that has been split due to a different quantity being requested (paid at 10 p per prescription where needed) and d denotes exceptional costs, such as delivery charges.

Although the actual cost more accurately reflects the total spend to the NHS, the addition of container payments and out-of-pocket expenses can affect the PPU, particularly where the prescribed quantities are small and are inexpensive, leading to multiple PPU figures for the same presentation. Using NIC avoids these multiple figures, and instead calculates the PPU based solely on NHS list price, providing a more consistent calculation, although with marginally overestimated savings.

## Policy implications and further research

We have identified significant opportunities for savings that arise because of complexities in the systems for pricing and dispensing medicines. Realising these savings requires that clinicians are given access to user-friendly tools that allow them to identify where their prescribing presents savings opportunities and helps them to identify the treatment with the lowest cost. We have recently launched a pilot version of such a tool at OpenPrescribing.net and will be monitoring user feedback and use statistics.

While determining which specific presentation (eg, branded generic) is cheapest is made far simpler using our tool, it still requires that clinicians modify their prescribing behaviour across many different drugs. It is possible that CCGs could consider using the tool as part of a GP prescribing incentive scheme or similar. However, it is still possible that relying on clinicians to make these switches in order to save money will place an additional time and cognitive burden on them, in contrast with 'always prescribing generically'. It is therefore our view that, rather than requiring individual doctors to achieve individual savings, much of the variation in PPU could be managed better through policy changes to address loopholes and oversights in the regulations around pricing and dispensing. While an extensive discussion is beyond the scope of this paper, much could also be achieved by changing the Drug Tariff price for a generic drug to more closely reflect the true price of currently prescribed low cost options. Addressing this variation in PPU centrally would protect more NHS funds and save clinicians' time. It would also allow GPs to continue to follow the best-practice recommendation to 'always prescribe generically'.

Of note, there is also considerable circularity in the management of the current drugs budget: CCGs are the primary gatekeepers for spending; but where they achieve savings on the drugs budget, this may be counteracted in the following year by modification of the prices of category M medicines, or other reimbursements to pharmacists, in a complex process intended to incentivise and preserve the presence of community pharmacies. In the current situation, where some CCGs have more information and capability to act on efficiency more than others, then more efficient CCGs will benefit disproportionately. Conversely, pharmacy contractors in areas with efficient prescribers will have their profits reduced disproportionately to other contractors. Therefore, any system which equalises access to price-saving opportunities would

increase equity across the country, both in terms of CCG funding and pharmacy reimbursement.

Lastly, we estimate that half of all the savings identified using this methodology are the result of purchasing or supply decisions which are out of the control of both the CCG and the prescriber. It seems peculiar that a pharmacist can choose to supply a more expensive pack size to fulfil a prescription that necessary, even where a prescriber has stated the lower pack size on the prescription. There is even greater discrepancy in the costs of drugs which are not listed in Part VIII of the Drug Tariff, including 'specials', imported medicines and those drugs which, despite being commonly available, are invoiced to the NHS at higher cost than expected (so-called 'NP8' medicines'). Given the level of variation of costs identified using this methodology, it would seem prudent for policy-makers to undertake a review of these issues.

## CONCLUSIONS

We have developed a new method to identify and enable large potential cost savings within NHS community prescribing. Given the current pressures on the NHS, it is vital that these potential savings are realised. Our tool enabling doctors to achieve these savings is now launched in pilot form. However, savings could potentially be achieved more simply through national policy change.

**Contributors** RC, SB, AJW, BG and HJC conceived and designed the study. SB and AJW collected and analysed the data with input from RC, HJC and BG. AJW drafted the manuscript. All authors contributed and approved the final manuscript. SB was lead engineer on the associated website resource with input from RC, AJW, BG, HJC and LF (who led on user testing). BG supervised the project and is guarantor. Lead engineer on the original OpenPrescribing tool was Anna Powell-Smith.

**Funding** This work was supported by the NIHR Biomedical Research Centre, Oxford; the Health Foundation grant (Unique Award Reference Number 7599) and by an National Institute for Health Research (NIHR) School of Primary Care Research (SPCR) grant ref number: 327.

**Competing interests** BG has received research funding from the Laura and John Arnold Foundation, the Wellcome Trust, the NHS National Institute for Health Research, the Health Foundation and WHO; he also receives personal income from speaking and writing for lay audiences on the misuse of science. AJW, HJC, SB, RC and LF are employed on BG's grant from the Health Foundation. RC reports personal fees as a paid member of an advisory board from Galen Pharmaceuticals, Martindale Pharma, Galderma (UK), ProStraken Group PLC, Menarini Farmaceutica Internazionale SRL, Stirling Anglian Pharmaceuticals , outside the submitted work; and RC is employed by a CCG to optimise prescribing.

**Patient consent** Detail has been removed from this case description/these case descriptions to ensure anonymity. The editors and reviewers have seen the detailed information available and are satisfied that the information backs up the case the authors are making.

**Provenance and peer review** Not commissioned; externally peer reviewed.

**Data sharing statement** All analytic data and code are available online at https://figshare.com/s/39a4301a29316bc86b35. All codes for the OpenPrescribing tool and the associated PPU tool are shared under an open license and is available on Github https://github.com/ebmdatalab/price-per-dose

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
