## [Reviewer comments · BMJ Open]

ARTICLE DETAILS

TITLE (PROVISIONAL)	A New Mechanism To Identify Cost Savings in English NHS Prescribing: Minimising "Price-Per-Unit", a Cross Sectional Study
AUTHORS	Croker, Richard; Walker, Alex; Bacon, Seb; Curtis, Helen; French, Lisa; Goldacre, Ben

VERSION 1 – REVIEW

REVIEWER	Prof Nick Bosanquet Imperial College UK.
REVIEW RETURNED	13-Nov-2017

GENERAL COMMENTS	1. Add a para on the background in increase in generic prescribing. This is regarded as having made it possible to increase the number of prescriptions far faster than the total spend. The study points to a next stage to deal with the unexpected effects by which there may be differences in the prices of generics and also differences in price across all available drugs. It would be useful to have some explanation of why this is the case. 2. Are there any quality problems which might arise from the new mechanism--- for example adherence might fall if the patient was used to one brand and had doubts about another label... 3. It would be useful to have a small diagram setting out the roles of the doctor/primary care team and the pharmacists in all of this. Apparently some of the saving could be reduced by independent actions of the pharmacist. 4. The paper could be shortened by 25%. It makes an important point--- but just the one point. As part of this the diagrams on pp14 and 15 could be dropped--- they add to public mystification rather than to enlightenment. The quoting of the main results is enough. 6. What are the incentives needed for practices and CCGs to use this system?
--

REVIEWER	Nick Hex York Health Economics Consortium, University of York, UK
REVIEW RETURNED	28-Nov-2017

GENERAL COMMENTS	I think this is an interesting and well argued paper. My only concern is that of the practical application of 'price per unit' switching at an individual practice level and adherence by individual GPs. I think this is addressed as a limitation and the authors are correct to say that the most practical implementation of this change could be achieved through national policy changes and mandating switching to cheapest presentations.
--

	I am happy to be convinced otherwise, and apologies if I have misinterpreted, but my feeling is that there would be considerable additional burden on practice staff to go from 36 actions per year for generic switching to 248 actions per year for price per unit switching. The average saving per action is not that different (£118 v £134) but I have two thoughts, a) how much time would be required to undertake these actions and what is the potential opportunity cost (b) how would they ensure that GPs would follow the advice issued when there would be effectively one action per working day. There may also be a behavioural issue whereby GPs simply prefer to use a particular brand or may feel it is not worth the hassle of explaining to patients why their drugs have apparently changed, in order to save a small amount of money. It may be worth exploring these issues a little further in the text, particularly what constitutes an 'action'. I like the way the authors have acknowledged and quantified other limitations such as the issue of rebates and pack sizes. The paper is well written and argued and I think a nationally implemented approach could work well and achieve considerable savings. One other minor point is that I wonder whether it is worth putting the aggregate average saving per practice in the abstract. I initially thought that each practice would only save £50 per month but the saving is much higher and only becomes apparent in the results section.
--	---

VERSION 1 – AUTHOR RESPONSE

Reviewer: 1

Reviewer Name: Prof Nick Bosanquet

Institution and Country: Imperial College, UK.

Competing Interests: None declared.

1. Add a para on the background in increase in generic prescribing. This is regarded as having made it possible to increase the number of prescriptions far faster than the total spend. The study points to a next stage to deal with the unexpected effects by which there may be differences in the prices of generics and also differences in price across all available drugs. It would be useful to have some explanation of why this is the case.

Response: Many thanks for all your comments. We have extended the discussion of the rise in generic prescribing in the introduction.

2. Are there any quality problems which might arise from the new mechanism--- for example adherence might fall if the patient was used to one brand and had doubts about another label...

Response: This is an important issue, we have added to the 'Achievability' section on this subject.

3. It would be useful to have a small diagram setting out the roles of the doctor/primary care team and the pharmacists in all of this. Apparently some of the saving could be reduced by independent actions of the pharmacist.

Response: We think that this is important, and while we have not added a diagram, we have edited Box 1 in order to better explain where the savings can be made.

4. The paper could be shortened by 25%. It makes an important point--- but just the one point. As part of this the diagrams on pp14 and 15 could be dropped--- they add to public mystification rather than to enlightenment. The quoting of the main results is enough.

Response: We have removed Figures 2 and 3, as well as any reference to them in the text. We have also slightly shortened the text in places without losing any of the important nuance being discussed.

6. What are the incentives needed for practices and CCGs to use this system?

Response: The underlying incentive for making these changes is to identify variation and save money, but in terms of mechanisms to incentivise use of the system, CCGs could consider using the tool as part of any GP prescribing incentive scheme or similar. We have added this to the discussion (pg. 21 last paragraph)

Reviewer: 2

Reviewer Name: Nick Hex

Institution and Country: York Health Economics Consortium, University of York, UK

Competing Interests: None declared

Comment: I think this is an interesting and well argued paper. My only concern is that of the practical application of 'price per unit' switching at an individual practice level and adherence by individual GPs. I think this is addressed as a limitation and the authors are correct to say that the most practical implementation of this change could be achieved through national policy changes and mandating switching to cheapest presentations.

Response: Many thanks for all your comments. We absolutely agree that national policy change is the best way to achieve the savings, but that in the meantime some of the biggest savings might be addressed using our tool.

Comment: I am happy to be convinced otherwise, and apologies if I have mis-interpreted, but my feeling is that there would be considerable additional burden on practice staff to go from 36 actions per year for generic switching to 248 actions per year for price per unit switching. The average saving per action is not that different (£118 v £134) but I have two thoughts, a) how much time would be required to undertake these actions and what is the potential opportunity cost (b) how would they ensure that GPs would follow the advice issued when there would be effectively one action per working day. There may also be a behavioural issue whereby GPs simply prefer to use a particular brand or may feel it is not worth the hassle of explaining to patients why their drugs have apparently changed, in order to save a small amount of money.

Response: We agree that the burden for making these savings currently falls on clinicians and CCG medicines optimisation teams and that this presents a potential additional cognitive burden in contrast with the simplicity of 'always prescribing generically'. We have expanded discussion of this matter in the discussion (pg. 21 last paragraph). We mention that switching between brands/generics can be difficult in the 'Achievability' section, and have now made this more explicit with an example.

Comment: It may be worth exploring these issues a little further in the text, particularly what constitutes an 'action'. I like the way the authors have acknowledged and quantified other limitations such as the issue of rebates and pack sizes. The paper is well written and argued and I think a nationally implemented approach could work well and achieve considerable savings.

Response: Our definition of an 'action' is stated on pg 9 in 'Describing variation'. We think that the expanded discussion on this now better describes this issue.

Comment: One other minor point is that I wonder whether it is worth putting the aggregate average saving per practice in the abstract. I initially thought that each practice would only save £50 per month but the saving is much higher and only becomes apparent in the results section.

Response: Thanks for the suggestion, we have added this to the abstract.

VERSION 2 – REVIEW

REVIEWER	Prof Nick Bosanquet Imperial College, UK
REVIEW RETURNED	12-Dec-2017

GENERAL COMMENTS	Much improved---very important contribution for freeing up funds for investment.
--

REVIEWER	Nick Hex York Health Economics Consortium,UK
REVIEW RETURNED	23-Dec-2017

GENERAL COMMENTS	I am happy with the responses of the authors and satisfied with the way they have addressed the issues I raised in the text.
--